# Yeast-Based Biosensors: Current Applications and New Developments

**DOI:** 10.3390/bios10050051

**Published:** 2020-05-13

**Authors:** Helene Martin-Yken

**Affiliations:** 1Institut National de Recherche pour l’Agriculture, l’Alimentation et l’Environnement (INRAE), UMR 792 Toulouse Biotechnology Institute (TBI), 31400 Toulouse, France; helene.martin@insa-toulouse.fr; Tel.: +689-89-52-31-88; 2Institut de Recherche pour le Développement (IRD), Faa’a, 98702 Tahiti, French Polynesia; 3Unite Mixte de Recherche n°241 Ecosystemes Insulaires et Oceaniens, Université de la Polynésie Française, Faa’a, 98702 Tahiti, French Polynesia; 4Laboratoire de Recherche sur les Biotoxines Marines, Institut Louis Malardé, Papeete, 98713 Tahiti, French Polynesia

**Keywords:** yeasts, biosensors, cell signaling, environmental contaminants, detection

## Abstract

Biosensors are regarded as a powerful tool to detect and monitor environmental contaminants, toxins, and, more generally, organic or chemical markers of potential threats to human health. They are basically composed of a sensor part made up of either live cells or biological active molecules coupled to a transducer/reporter technological element. Whole-cells biosensors may be based on animal tissues, bacteria, or eukaryotic microorganisms such as yeasts and microalgae. Although very resistant to adverse environmental conditions, yeasts can sense and respond to a wide variety of stimuli. As eukaryotes, they also constitute excellent cellular models to detect chemicals and organic contaminants that are harmful to animals. For these reasons, combined with their ease of culture and genetic modification, yeasts have been commonly used as biological elements of biosensors since the 1970s. This review aims first at giving a survey on the different types of yeast-based biosensors developed for the environmental and medical domains. We then present the technological developments currently undertaken by academic and corporate scientists to further drive yeasts biosensors into a new era where the biological element is optimized in a tailor-made fashion by in silico design and where the output signals can be recorded or followed on a smartphone.

## 1. Introduction

Biosensors are among the most sensitive screening methods used to detect harmful chemicals and pollutants. According to their classically accepted definition, biosensors are composed of a biological recognition element able to sense or interact with the target molecules coupled to a physicochemical transducer and a microelectronic processor functioning as amplifiers and converters of the biological response into a measurable/numerical signal [1]. In these devices, the biological elements can be antibodies, specific proteins such as cell receptors or enzymes, nucleic acids, organelles, tissues, microorganisms, or whole cells. In this last category the whole cells, either eukaryotic or prokaryotic, are used as reporters with the advantage of combining both the biological receptor and the transducer elements in one [2]. Cells or micro-organisms used as whole-cell biosensors can be genetically modified in various ways in order to increase their sensitivity or to incorporate different reporter and transducer capacity [3,4]. 

Most biosensors have been developed based on bacterial cells; however, eukaryotic cellular models present several relevant advantages. Among them, yeasts are of special interest, given their resistance to harsh environmental conditions, their successful long-term relationship with humans [5], and the fact that they are very well known at both technological and genetic levels. As eukaryotic organisms, yeasts share most cellular features and molecular mechanisms with our mammalian cells and notably several signaling pathways which are of great relevance towards sensing and responding to environmental stimuli. This high level of conservation has long driven scientists to use yeasts as model eukaryotes for the study of a wide range of cellular biology processes [6]. The best studied among yeasts species, *Saccharomyces cerevisiae* (also known as bakers’ yeast) was the first eukaryotic organism whose genome was entirely sequenced [7] and is remarkably easy to modify genetically. Yeasts grow fast on inexpensive culture medium. They are very robust organisms that tolerate a wide range of temperatures, and they can be frozen or dehydrated for storage and transportation purposes. The combination of these elements (conservation of eukaryotic pathways and cellular mechanisms) with the practical aspects such as safety and easiness to cultivate, transport, and conserve yeast cells makes them an extremely interesting choice of biological model for the development of biosensors [5]. In addition, from an ethical point of view, the choice of yeast cells also allows using non-animal models to determine the potentially toxic effects of very diverse compounds or inversely to screen for therapeutic molecules (see below). Bioassays and biosensors based on yeast cells have been emerging over the years and are actually in use in various domains of application. In this review, we describe the different types of biosensors based on yeast cells with a special focus on environmental and medical applications; this distinction, however, is sometime hard to make and can appear arbitrary since what makes environmental contaminants harmful to Man or wild-life is precisely their effects on health. Hence, some biosensors or yeast-based screens described in this review can be considered as relevant for both of these application domains. Figure 1 depicts the general principle of yeast-based biosensors, with the possible inputs, the sensing and detection elements, and the desired output response. 

First, yeast cells either native or modified to constitutively produce luminescence can be used as non-specific reporter systems to monitor the toxicity toward eukaryotic cells of compounds found or used in food, the environment, building materials, cosmetology, drug design, etc. [8]. However, toxic compounds vary greatly in their cytotoxicity mechanisms; some are non-toxic for yeast cells while they may be toxic to human cells and tissues. In addition, yeasts have developed highly efficient detoxifications mechanisms and efflux pumps such as the pleiotropic drug resistance (PDR) family of ATP-binding cassette (ABC) transporters, which are able to export from the cell a broad range of chemically distinct molecules resulting in multidrug resistance [9]. Hence, using yeast cells to assess non-specific toxicity toward mammals remains tricky and demands a very careful optimization of the incubation conditions and duration. In that respect, genetically modified yeast strains have been designed by several different labs over the last few decades in order to detect specific molecules or families of compounds. Yeast-based sensing technology has thus evolved from using the natural potential of yeast cells, such as their sensitivity to toxic molecules or their ability to metabolize organic compounds and simply following their growth, toward the design of more and more complex genetically modified strains. Notably, many biosensors have been constructed by integrating heterologous genes in yeast cells, conferring them new recognition capabilities. These exogenous sensors proteins can be coupled directly or indirectly to transcription factors that, in turn, activate a reporter gene, either metabolic or driving a signal that can be easily followed by colorimetry, fluorescence, luminescence, amperometry, etc. Such approaches have been used by yeast scientists worldwide to design biosensors for a wide range of applications (see below, Section 2). However, several other smart sensing mechanisms have also been developed for specific purposes, such as using the yeast genetic recombination frequency to assess the presence of genotoxic compounds or radiation. Yeast-based sensing technology is indeed a field in constant evolution, and increasingly sophisticated mechanisms are currently being designed. Moreover, the rise of synthetic biology combined with computer-assisted structural biology is opening exciting future prospects (see below, Section 3). 

## 2. Current Applications

Table 1 collects in a non-exhaustive attempt the main types of yeast-based biosensors that have been developed throughout the last decades either for monitoring environmental pollutants or dedicated to the medical domain. As can be seen in Table 1, *Saccharomyces cerevisiae* is the most frequently chosen host due to the high number of genetic tools available for this background. However, several other yeast species such as *Hansenula polymorpha, Kluyveromyces fragilis*, *K. marxianus, Pichia pastoris*, and *Arxula adenivorans* appear to be better models for specific sensing purposes. These yeasts cells can be used as such (“per se”) as the sensing element of a biosensor. In this case, the output signal is represented by a change in the cell’s metabolism or viability. Alternatively, yeast cells can be genetically modified to express heterologous or chimeric proteins able to specifically sense or interact with the molecules of interest and allow a signaling mechanism to take place. These sensing proteins are either membrane-bound receptors or channels, intracellular signaling pathway members, or even direct transcription factors. Figure 2 illustrates these different key features of yeasts cells used as sensitive elements in current biosensors. 

### 2.1. Environment 

Biosensors based on yeast cells have been extensively developed in the aim of detecting environmental pollutants as recently reviewed by Jarque and colleagues [42]. Chemical compounds potentially harmful for human health can now be found everywhere in the air, water, and soil, even in remote areas previously considered preserved from any pollution caused by humans. Among the different approaches commonly used for the detection of environmental pollutants, yeast-based methods have often been sought, since they present advantages similar to prokaryotic assays but are more representative of higher organisms. Several yeast-based biosensors are already used routinely as convenient tools either to evaluate the toxicity of pollutants on eukaryotic cells or monitor the level of contamination of environmental samples. Some, however, still need improvement regarding their specific limitations in terms of ease of use, shelf conservation, applicability, and potential use in high-throughputs formats. The environmental pollutants targeted by these biosensors comprise metals, endocrine disruptants, genotoxins, and cytotoxins as well as a large and more confused group of “biodegradable organics” considered of increasing concern to aquatic environments. The amount of these organic pollutants in water samples is most often assessed by quantification of the biochemical oxygen demand (BOD). BOD corresponds to the amount of dissolved oxygen needed by aerobic biological organisms to degrade the organic material in the sample at certain temperature over a specific time period (generally five days at 20 °C). BOD can be measured using bacteria, but yeast species that have broad substrate ranges are also successfully used for this purpose. For example, Hikuma and coworkers used *Trichosporon cutaneum* as early as 1979 to construct a rapid BOD sensor [29]. Middelhoven and colleagues selected a different yeast, *Arxula adeninivorans*, for its ability to catabolize many substrates including various nitrogenous and aromatic compounds [43]. BOD determination is indeed a major application of yeasts as the biological element of biosensors, and countless variations have been developed with different yeasts to target different organic compounds, but also with different detection systems gradually improving over the years [31,44,45,46]. Notably, a new automated chemiluminescence method using sequential injection analysis (SIA) has recently been developed [47]. Based on a relatively simple technology (the redox reaction between a quinone and *S. cerevisiae* in the presence of organic substances), this test appears as an economic and high-throughput screening bioassay alternative to conventional methods to assess BOD in environmental samples. Further developments are ongoing in several labs with amazing detection technologies coupled to up-to date devices. 

**Endocrine disruptants**: The potential estrogenic activity of pollutants has been a major target addressed by yeast bio-assays, starting with an *S. cerevisiae* –based screen developed by Routledge and Sumpter to assess the estrogenic activity of surfactants and their degradation products [11]. This historical assay called “YES” (for yeast estrogen screen) relied on the construction of an estrogen-inducible expression system in yeast, with the human estrogen receptor gene integrated into the yeast main genome and estrogen response elements placed on a plasmid allowing expression of the reporter gene Lac-Z upon activation of the receptor. Lac-Z encoded β-galactosidase expression level was then followed by colorimetry. Although of bacterial origin, the *E. coli* Lac-Z gene is correctly expressed and transcribed in yeast, yielding a robust enzyme with a remarkably stable activity [48]. This gene has thus been used as a reporter of gene expression in yeast for many years for a large number of promoters [49]. For a nice sum up of the most common reporter genes used in biosensors and their characteristics, see Gutiérrez, Amaro, and Martín-González [1] and Table 1 therein. A different recombinant yeast assay targeting estrogens was developed later by Garcia-Reyero and colleagues [12] also by expressing the human estrogen hormone receptor in yeast and β-galactosidase expression but followed this time by a fluorescent enzymatic dosage. Shortly after, other recombinant yeast strains were constructed that expressed the human estrogen receptor and ß-Galactosidase (ßGal), Luciferase (Luc), or yeast Enhanced Green Fluorescence Protein (yEGFP) as a reporter proteins with the purpose of routine screening of estrogen activity in complex matrices such as agricultural products. Of these, the yEGFP proved to be the best output as the assay could be performed completely in 96 well plates within 4 h [13]. The suitability of these three recombinant yeast-based assays as a pre-screening tool for monitoring estrogenic activities in water samples was evaluated in an inter-laboratory study [50]. No significant difference was found between the performances of these tests, which also showed a good correlation with expected values from chemical analysis by LC-MS/MS. Several laboratories designed variants of these assays [51,52,53], while others went instead looking for more specific tests. Indeed, among these environmental endocrine disrupters, some but not all displayed androgenic and anti-androgenic activities. Yeast detection screens were constructed based on the yeast two-hybrid system for protein interactions, in which androgens, but not other hormones, strongly stimulated the β-galactosidase activity in a dose-dependent manner [15,16]. Moreover, the hunt for specificity has led researchers to perform directed evolution on the human receptor used in their yeast test. A sensor able to specifically identify bisphenol A (BPA) and distinguish it from other estrogenic compounds has been obtained this way [54].

One important advantage of yeast-based systems to monitor estrogens compared to mammalian-based tests is the absence of side effects potentially altering the results. Indeed, mammalian cells culture medium can easily be contaminated by interfering molecules (for example, steroids) present in the fetal bovine serum. Numerous studies have shown that yeast systems are valuable for screening hormonal substances [55], and yeast-based bioassays targeting estrogens and androgens are in use today to investigate the presence and endocrine activities of pesticides in environmental samples such as wastewater effluents [56,57], fish oils [58], etc. [59]. One of them, the “EstraMonitor,” uses cells of *Arxula adeninivorans* in an automated system that allows for semi-online continuous monitoring of estrogenic compounds in wastewater samples [14]. The same team also developed biosensors based on a similarly genetically modified *A. adeninivorans* yeast strain for the detection of pharmaceuticals such as omeprazole, lansoprazole, etc., and for glucocorticoids including cortisol, corticosterone, and prednisolone [17,18].

**Heavy metals:** Among inorganic pollutants, heavy metals are considered as a very serious human thread and a priority environmental concern. They are particularly good targets for biosensor development because many types of cells are highly sensitive to metals and have evolved pathways to rapidly and intensely respond to metal stress. These cellular responses mainly rely on metallic ion chelating molecules, such as glutathione, phytochelatins, and metallothioneins. The promoters of the genes involved in these cellular pathways are strongly induced by heavy metals and can be used as key elements of specific highly sensitive biosensors. Alternatively, promoters of genes regulating the expression of antioxidant enzymes such as superoxide dismutases, catalases, glutathione peroxidases, etc. can be used many of them are also strongly induced by metals. For an updated review, see Gutiérrez, Amaro, and Martín-González [1]. Currently, copper ions can be detected by yeast cells bearing plasmids with the *CUP1* promotor fused to either LacZ gene [22], GFP for a fluorescent detection [23], luciferase [24], or *ADE2* gene for a colorimetric output signal [25]. In an attempt to target cadmium ions, a *Hansenula polymorpha HSEO1* gene promoter has been isolated and used to direct specific expression of GFP upon exposure to Cd in a dose-dependent manner, with Cd detection ranging from 1 μM to 900 μM, hence proving its interest for whole cell biosensor, although it responds to arsenic as well to a lesser extent [26]. Cu^2+^ and Cd^2+^ heavy metal ions are also efficiently detected by a whole-cell electrochemical biosensor based on mixed microbial consortium containing *S. cerevisiae* together with *E. coli* and *B. subtilis* bacterial species, an innovative [60]. Among heavy metals, exposure to mercury—particularly in the form of methylmercury—is acknowledged to produce significant adverse neurological and other health effects, with harmful effects most evident on unborn children and infants (Exposure to mercury: a major public health concern. Geneva, World Health Organization; www.mercuryconvention.org). Yeast cells are capable of sensing and accumulating methylmercury, hence they constitute a choice cellular model for the design of biosensors targeting this pledge [61].

**Marine toxins:** One of the major risks to human health caused by climate change is the associated challenge of ocean acidification and temperature rise, which appear to be changing the distribution and frequency of harmful algal blooms (HABs) [34]. In HABs, micro-algae at the base of the marine food chain produce toxins—such as okadaic acid, brevetoxins, ciguatoxins, pectenotoxin, yessotoxins, etc.—that can severely affect human health, with effects ranging from digestive troubles to skin irritations, respiratory and neurological diseases, and even death. Recently, Richter and Fidler reported the development of recombinant *S. cerevisiae* strains expressing tunicate VDR/PXRa receptors of *C. intestinalis* and *B. schlosseri* as fusion proteins combined with the GAL4-DNA binding domain and a generic transcription activation domain easily assayed by lacZ reporter gene. This set of yeast strains enables the detection of different ligands for these tunicate receptors, including carbamazepine and bisphenol-A, as well as more structurally complex marine biotoxins such as okadaic acid, pectenotoxin-11, and portimine. These recombinant yeasts can be used in high-throughput robust and inexpensive screens for microalgal biotoxins and novel marine bioactive chemicals [27]. Recombinant *S. cerevisiae* strains able to detect ciguatoxins, the potent neurotoxins produced by *Gambierdiscus* and *Fukuoya* spp., have also been recently developed [28]. 

**Mycotoxins:** As the most common contaminants of food and feed worldwide, mycotoxins represent a major threat for human and animal health worldwide. The term mycotoxins comprises a wide range of toxic compounds produced by different types of fungi, mainly from the *Aspergillus, Penicillium*, and *Fusarium* genera. Upon proliferation of these fungi, mycotoxins enter the food chain through contaminated food and feed crops (e.g., cereals, milk) and severely affect human and animal health [35]. Some mycotoxins are among the most powerful known inductors of cancer and mutations, as well as estrogenic, gastrointestinal, and kidney disorders. Others have immunosuppressive effects, thereby reducing resistance to infectious disease [62]. Their mechanism of action is believed to be linked to oxidative stress [63]. As early as 1984, an assay was developed to detect T2 toxin using the yeast *Kluyveromyces fragilis*, which was also sensitive to other trichothecenes such as verrucarin A [19]. This assay failed to detect Aflatoxin B1 and zearalanone but was nevertheless further redesigned as a colorimetric bioassay to detect trichothecene mycotoxins using inhibition of beta-galactosidase activity in the yeast *Kluyveromyces marxianus* [20]. Zearalenone (ZON) is a non-steroidal estrogenic mycotoxin produced by *Fusarium* spp. that can be found in cereals and derived food products. To design a sensitive and cheap assay to monitor ZON levels in grains, a *S. cerevisiae* strain unable to grow unless the expressed human estrogen receptor is activated has been engineered. This strain allows the qualitative detection and quantification of total estrogenic activity in cereal extracts, with a sensitivity suitable for low-cost monitoring of ZON and other estrogenic compounds [21]. 

Finally, in order to detect a wider range of compounds present in the environment that may be toxic to eukaryotic cells, less specific yeast-based biosensors have also been developed, notably by Hollis and colleagues [10]. Remarkably, this biosensor tested on herbicides and heavy metals manages to detect toxicity of compounds that are not revealed by prokaryotic biosensors. Similarly, the vacuolar metabolism of yeast serves as a biomarker for the detection of heavy metals, pesticides, and toxic pharmaceuticals in a recently developed test based on the oxidative stress generated by these compounds [64].

### 2.2. Medical Domain/Health 

Biosensors offer a promising solution to improve “point-of-care” (POC) diagnostic worldwide. Yeasts have historically proven to be an excellent model to study mammalian diseases in a simpler organism [65] and are thus very useful to develop biosensors directly relevant to human health. For example, a cellular assay has been developed in the yeast *S. cerevisiae* that detects a characteristic protease activity of the human cytomegalovirus (HCMV) [41]. 

#### 2.2.1. Detection of Pathogens

In addition to viruses, yeast biosensors can be used to detect a wide range of microbial pathogens including fungal pathogens. In a remarkable work, Ostrov and colleagues developed a highly specific colorimetric assay based on *S. cerevisiae* for detection of pathogen-derived peptides. They integrated G protein coupled receptors (GPCRs) to a visible and reagent-free lycopene readout and developed yeast strains which can differentially detect major human, plant, and food fungal pathogens with a nano-molar sensitivity. First developed for the detection of the human fungal pathogen *Candida albicans* it was then expanded to nine other major human, agricultural, and food spoilage pathogens: *Candida glabrata, Paracoccidioides brasiliensis, Histoplasma capsulatum, Lodderomyces elongisporus, Botrytis cinerea, Fusarium graminearum, Magnaporthe oryzae, Zygosaccharomyces bailii*, and *Zygosaccharomyces rouxii.* In addition, this assay has been further optimized into a one-step rapid dipstick prototype suitable for complex samples such as blood, urine, or soil [33].

**Carcinogens:** Yeasts and fungal cells have already been considered as relevant tools to investigate carcinogens for three decades [66]. The yeast DEL assay designed by Brennan and Schiestl to detect carcinogens is a simple and rapid method to study the effects of various DNA-damaging treatments on the frequency of deletion-recombination [67,68]. It shows a high sensitivity and specificity toward carcinogens poorly detected by bacterial mutagenicity and other short-term genotoxicity assays and has been developed as a high-throughput screen [69]. With a different concept, a yeast-based biosensor using a *HUG1* promoter-*GFP* reporter has been constructed [36]. It allows the detection of a wide variety of genotoxic compounds, including alkylating and oxidative agents, a ribonucleotide reductase inhibitor, a UV mimetic agent, an agent that causes double strand breaks, a topoisomerase I inhibitor, and even ionizing radiations at various doses. In addition to carcinogens, numerous chemicals such as polycyclic aromatic hydrocarbon and mycotoxins are pro-carcinogens: i.e., they become carcinogenic only after they have been bio-activated by cellular metabolic processes. Ngoc Bui and colleagues designed a yeast-based biosensor able to determine and evaluate pro-carcinogens presents in environmental samples [37]. 

#### 2.2.2. Drug Discovery

Moreover, in addition to the direct biosensing of toxic compounds, yeasts bioassays directly useful to medical/health research have also been designed in distinct orientations such as to develop large scale screening methods for new drugs discovery (see below) including for mitochondrial dysfunctions [70]. Toxins trafficking can also be efficiently deciphered with yeast cells as tools. This last application type has been successfully applied to ricin, the worldwide famous toxin used as lethal poison and biological warfare agent by spies and terrorists [71]. Regarding drug discovery applications, several yeast-based bioassays have been developed to be used as screens for a wide variety of threats to human health such as viruses, cancer, parasites, or prion-related diseases. A few examples are listed below. 

**Anticancer treatments:** Human matrix metalloproteinases (MMPs) and their different malfunctions are implicated in severe diseases including cardiovascular troubles and cancer development. Inhibitors of specific matrix metalloproteinases are therefore potential therapeutic targets and their search is one of the leads in oncology research. Diehl and colleagues developed a recombinant *Pichia pastoris* yeast strain that expresses biologically active human MMPs at the cell surface allowing to screen for MMPs [39]. Another screen based on the model yeast *S. cerevisiae* allows the identification of inhibitors of phosphatidylinositol 3-kinase (PI3K) [38]. Since the PI3K pathway is involved in tumorigenesis but also in several other common pathologies, including autoimmune and cardiovascular diseases, this robust bioassay (applicable to large-scale HTS) is of great interest for medical research not restricted to oncology. 

**Anti-protozoans:** Malaria is a life-threatening disease caused by *Plasmodium* parasites (notably *P. falciparum*) transmitted to people through mosquito bites. According to the World Health Organization, nearly half of the world’s population is at risk of malaria, with a number of victims as high as 219 million cases of malaria and 435,000 deaths in 2017. Currently, due to the appearance of resistance to the previous treatments, Artemisinin is the only really efficient drug in use used to treat malaria. A bioassay has been developed in *Saccharomyces cerevisiae* to screen for compounds with artemisinin-like activities to help both the search for new treatments and a better understanding of the drug’s mode of action [40]. 

**Against prions:** Since the 1990s, budding yeast has proven an excellent model for the study of prions [72,73]. Using the conservation of the biochemical mechanisms controlling prions formation and maintenance between yeasts and mammals, Bach and colleagues designed a rapid two-steps yeast-based assay to isolate drugs active against mammalian prions [74,75]. Their method was validated as an efficient high-throughput screening approach to identify prion inhibitors and allowed the isolation of a new class of compounds, the kastellpaolitines, able to promote mammalian prion clearance. Remarkably, this screen for prions pharmacological inhibitors might also lead to isolate molecules active on other amyloid-based pathologies like Alzheimer’s, Parkinson’s, and Huntington diseases. 

**Search for new antifungals :** In 2005, Heinisch described a set of genetic constructs applicable in the investigation of stress signaling pathways and protein kinase inhibitors [76]. These constructs combined to different high-throughput techniques can be used to detect toxic compounds affecting PKC1 signaling in eukaryotes or simply to screen for new antifungal drugs.

## 3. Current and Future Developments

With the ever-increasing rhythm of human activities worldwide, the degradation of environmental conditions is happening at an accelerated pace. Air, soil, fresh and marine waters, and even remote locations that were earlier considered as pristine are now contaminated by a variety of pollutants including potentially toxic elements such as pesticides, toxins and endocrine disrupting molecules, chemicals, and heavy metals. Hence, environmental monitoring is now a priority for human health, and biosensors are particularly well adapted to this field since they are generally cost-effective, in situ, convenient, and real-time analytical techniques. Moreover, the recent rise of nanotechnologies allows the design of fast and smart biosensors containing nanomaterials or nanocomposites with improved analytical performances. The major current advances in yeast-based biosensor development are described below and depicted on Figure 3.

### 3.1. Metabolic Biosensors 

More and more synthetic biology processes are developed using yeasts as cell factories. They are however often delayed by the lack of screening procedures, which are lagging far behind and generally rely on classical analytical methods. Hence, private and academic research teams are trying to design intracellular metabolic biosensors allowing real-time monitoring of either the end product or a metabolic intermediate of interest [77,78]. The development of such biosensors is often based on transcriptional mechanisms and typically starts by exploring natural biodiversity genetic resources. Different engineering approaches are then used to fulfill specific industrial biotechnology applications [79]. Implementation in *S. cerevisiae* of allosterically regulated transcription factors from other species as metabolite biosensors appears as one of the possible strategies. The design of such biosensors relies on systematic engineering and can use small-molecule binding transcriptional activators of various origins including prokaryotic. This was notably demonstrated by Skjoedt et al., who showed that several activators from different bacterial species are indeed able to function as allosterically regulated transcription factors in *S. cerevisiae*, thus creating a biosensor resource useful for future biotechnological [80,81]. Following this demonstration, they further developed a synthetic selection system that couples the concentration of muconic acid (a plastic precursor) to cellular fitness using the prokaryotic transcriptional regulator BenM upstream of the Kan^r^ antibiotic resistance gene [82]. This sensor selector can be used to selectively enrich the best-producing cells from large libraries and isolate high-performance strains. 

To follow in real time metabolite levels in cultures is a key target. Converting these levels to fluorescence signals allows the monitoring of intracellular compound concentrations in living cells. Such a sensor has been constructed in *S. cerevisiae* for malonyl-CoA [83]. Zhang and colleagues designed a transcription factor-based NADPH/NADP+ redox biosensor in S. cerevisiae that can be used to monitor oxidative stress and changes in NADPH/NADP+ ratios. Coupled with dosage-sensitive genes (DSGs) expression, it constitutes a sensor-selector tool for synthetic selection of cells with higher NADPH/NADP+ ratios within a mixed cell population [84]. 

### 3.2. Multi-Strain Biosensors 

Taking advantage of specific characteristics not only of a single yeast strain but of several different microbial strains with complementary abilities is a very exciting lead that is already in use for environmental pollution assessment. Although it significantly increases the biosensor performance by broadening the range of detected substrates, it also poses the problem of maintaining the ideal balance between the different strains used that are in this case not restricted to yeast but can include a mix of yeasts and bacterial strains with different growth rates. This is an issue that can be solved by using native biofilms cultured in controlled conditions [85] or simply by immobilizing the cells in a convenient matrix. This last strategy was recently used by Yudina and colleagues to develop a biosensor to assess wastewater contaminations based on a co-culture of three distinct yeasts (*Pichia angusta, Arxula adeninivorans*, and *Debaryomyces hansenii*) and a bacteria (*Gluconobacter oxydans*) immobilized in a matrix of N-vinylpyrrolidone-modified poly(vinyl alcohol) [30]. Their results have established the potential of such co-cultures as the biological element of performant biosensor prototypes for broad applications. Similarly, an electrochemical biosensor for multi-pollutants toxicological analysis has been constructed by co-immobilizing mixed strains of *Escherichia coli, Bacillus subtilis*, and *S. cerevisiae* [60]. 

### 3.3. Technological Developments 

The rise of nanotechnologies within the last few years has already led to strong alterations in the conception of biosensor transduction systems, and it will necessarily continue to spectacularly enhance their diagnostic capability [86]. For example, the miniaturization of biosensors on microfluidic stages allows for multiplexing analysis with several biosensing strains used on a single chip detecting multiple contaminants in one step. Very recently, a variant of the multi-reporter yeast biosensor for the detection of genotoxic compounds has been developed. It is based on the recombination frequency measurement and uses fluorescence-activated cell sorting (FACS) with a multi-mode reader in 96-well black microplates suitable for high-throughput analysis [87]. In the field of estrogens contaminations monitoring, new developments of the bioluminescent yeast-estrogen screen (nanoYES) have led to a portable platform with a low-cost compact camera as a light detector and wireless connectivity, enabling a rapid and quantitative evaluation of total estrogenic activity in small sample volumes [88]. This newly developed and highly sensitive yeast biosensor can be connected wireless with any smartphone model for on-site analysis of endocrine disruptors applications in various type of samples [89]. These technological developments in detection and results analysis are regarded by healthcare providers as promising perspectives toward simplifying, standardizing, and automating biosensor-based diagnostic techniques, as they allow combining high precision and sensitivity with the connectivity and computational power of smartphones. Hence, clinical smartphone diagnostics methods are a rapidly emerging field whose area of applications ranges from hematology to rapid infectious disease diagnostics and digital pathology [90].

However, those technological developments for detection are only one aspect of the future trends. The most promising improvements to come for the development of new biosensors may in fact be artificial proteins and protein complexes created as desired by design. For example, regarding point of care diagnostic applications, an ultra-low-cost renewable bio-brick has been successfully designed and developed. Yeast cells can be genetically modified to display single-chain variable fragment antibodies on their surfaces as well as gold-binding peptide allowing a simple one step enrichment and surface functionalization. This strategy has been used with yeast cells displaying an antibody fragment directed to a *Salmonella* outer membrane protein, and a complete electrochemical detection assay could be performed with nanomolar detection limits [32]. The same versatile bio-brick approach has also been applied to construct an integrated diagnostic platform that combines sensing of Hepatitis C virus core antigen with a connected signal acquisition/processing through a smartphone-based potentiostat [91]. Hence, coupled to low-cost detection, this versatile approach constitutes a promising and exciting strategy which can be developed with virtually any type of pathogen provided that efficient antibodies have been developed. The approach of Ostrov and colleagues using GPCRs is equally versatile and can be adapted to detect target molecules that we have not yet thought of [33]. Similarly, Shaw and colleagues engineered a modified *S. cerevisiae* cell as a platform for biosensors construction by rational tuning of GPCR signaling [92,93]. 

### 3.4. In Silico Design

Moreover, synthetic structural biology methods that allow the design and construction of such chimera are constantly improving and becoming easier thanks to computer-assisted rational and computational approaches [94,95]. Such an approach allowed the design of a new synthetic biosensor capable of sensing and reporting the intracellular level of 4-hydroxybenzoic acid (pHBA) an important industrial precursor of muconic acid in *S. cerevisiae* [96]. Scientists are already capable now of designing entirely new proteins “from scratch” which perform new biological activities in vivo [97]. A strategy frequently adopted for biosensors design is to construct switch-like hybrid proteins where the binding of the molecule of interest induces conformational changes modulating the function (possibly enzymatic activity or luminescence) of the other domain. So far, the domain responsible for binding the target molecule was found in nature and sometimes improved by random or directed mutagenesis. However, it is now theoretically possible to design new proteins able to bind with high affinity to specific target molecules whose presence we wish to reveal [94]. Combining such a domain with a measurable activity turned on or off upon binding to the target molecule will allow for the design of de novo biosensors, with the choice of the cellular model and the output signal. Coherently with its position as the first microorganism domesticated and used by mankind, *S. cerevisiae* will soon become the first eukaryote redesigned in silico through de novo synthesis, reshuffling, and genome editing, an achievement currently undertaken by a large international consortium called the “Synthetic Yeast Genome (Sc2.0) Project” (http://syntheticyeast.org/). In this project, the entire yeast genome is being redesigned to allow it to be extensively rearranged on demand for various biotechnology applications thanks to a versatile genome-shuffling system [98]. Hence, a whole new era is opening up for the development of yeast-based biosensors.

## 4. Conclusions

The variety of cellular mechanisms exploited by researchers to design biosensors in yeast cells is remarkably wide. Sensing mechanisms have been sought for not just in the fungal kingdom but among the whole natural biodiversity—or at least the part of this diversity for which scientific knowledge was available. The advent of “omics” induced a major change in this search since it is now possible to look for specific activities (e.g., sensing or binding a target compound) in a mixed population of microorganisms such as a microbiota, even if most members of this microbial community are not identified individually and sometimes cannot be cultured as isolated clones. Moreover, the combined progresses of in silico modelling of protein structures and gene synthesis open another gate. The promise of these technologies for scientists is to no longer be limited to already existing, naturally occurring proteins. In the not too distant future, it could be possible, if not easy, to design new proteins with binding domains tailor-made for the target molecule we wish to detect. Even though some of these potential revolutions in the design of the biological element of biosensors are still far away, the technical progress on the other side are already functional and in constant evolution. Smartphone connectivity is only one of them, but an interesting one because it could sensibly ease the access to results of environmental monitoring from any location. In addition, it could also lead to the involvement of people outside the scientific community *strico sensu* into widened surveys and participative science projects. To conclude, biosensors based on yeast cells have been developed using several distinct sensing mechanisms, and many are already in use to detect and sense a wide variety of contaminants. In the near future, progress of both synthesis biology and nanotechnologies should prompt the development of many more of these convenient and easy to use detection and monitoring tools. 

## Figures and Tables

**Figure 1 biosensors-10-00051-f001:**
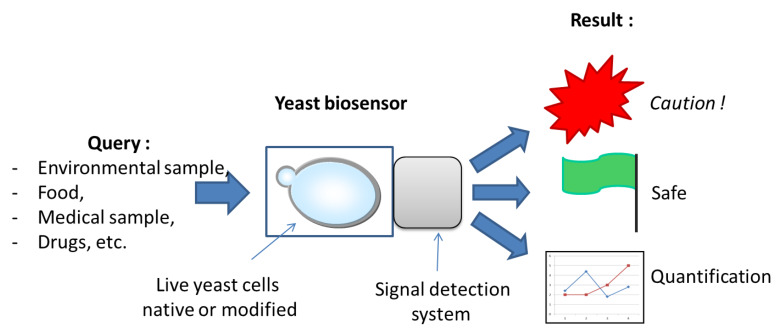
General scheme of a yeast biosensor’s purpose and functioning. Different possible inputs appear on the left, in a non-exhaustive list. Live yeast cells are represented by a budding yeast shape inside of a supporting structure that is coupled to the signal detection system. Three main outputs are generally sought after by designers and users: either a “yes/no” answer in case a threshold level of the target molecule(s) exists, or a quantification value when needed and possible.

**Figure 2 biosensors-10-00051-f002:**
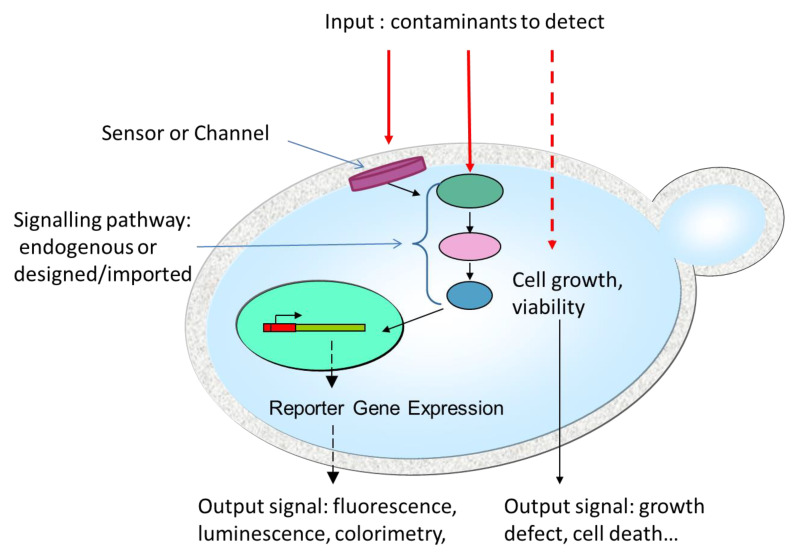
Key features of yeasts cells as biosensor sensitive elements. While some environmental contaminants can penetrate into the yeast cell and directly affect cell growth or viability, others (and particularly the biggest molecules) are retained outside the cells. These molecules can, however, be detected through membrane sensor proteins or channels that transmit signals to intracellular elements comprising signaling cascades and transcription factors. The outputs can be either a direct effect on cellular growth or viability or more indirect signals mediated through enzymatic activity and often, but not necessarily, gene expression.

**Figure 3 biosensors-10-00051-f003:**
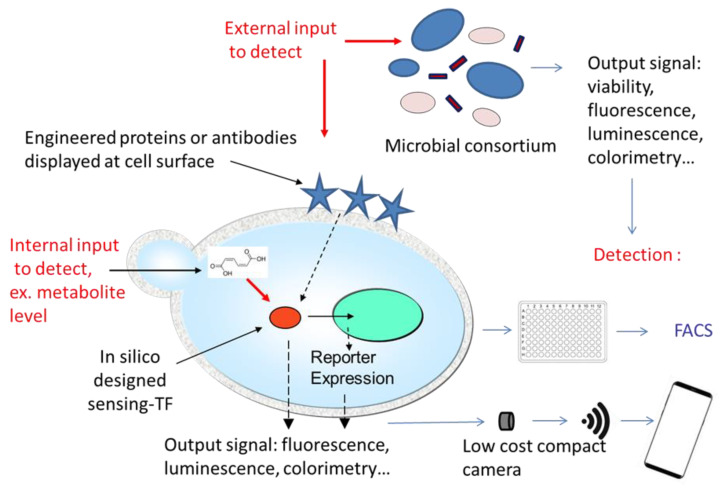
New developments and current research strategies in yeast-based biosensors. Remarkable new developments notably include multi-strain biosensors based on microbial consortia stabilized in a supporting matrix and metabolic reporters allowing users to monitor the metabolic state of yeast cells in a fermenter and hence make possible the progression of a biosynthetic process in real time. Moreover, nanotechnologies have revolutionized the detection systems by allowing both miniaturization and wireless transmission of data. Finally, in silico design of new binding partners for the target molecules and even unnatural amino acids use open unlimited options of new yeasts biosensors in the years/decades to come.

**Table 1 biosensors-10-00051-t001:** Different types of biosensors developed based on yeast cells. The upper part of the table summarizes yeast biosensors targeting pollutants and other environmental contaminants, while the lower part of the table contains bioassays developed for the medical domain to detect pathogens and carcinogens compounds or to be used as screening methods to help medical research (for example, the search for new drugs). The third column indicates the type of detection “Yes/No” or “Quantification,” as well as the limit of detection (LoD) or EC_50_ when such information was available. However, biosensors sensitivities vary significantly for different compounds and conditions; precise numeric values for specific molecules should be sought for in the original publications cited in the last column.

Detected Coumponds	Yeast Species	Type of Response (LoD or EC_50_ if Available)	Detection (Reporter Gene)	References
**Environment:**				
Coumpounds "toxic to eukaryotic cells" (all types)	*Saccharomyces cerevisiae*	Yes/No	Luminescence (Luc), viability decrease.	(Hollis et al., 2000) [10]
**Estrogenic coumponds (Endocrine Disruptors)**				
	*Saccharomyces cerevisiae*	Yes/No (2 ng/L)	Colorimetry (LacZ)	(Routledge and Sumpter, 1996) [11]
	*Saccharomyces cerevisiae*	Quantification (20 ng/L)	Fluorescence (LacZ)	(García-Reyero et al., 2001) [12]
	*Saccharomyces cerevisiae*	Quantification (0.4 nM)	Fluorescence (yEGFP)	(Bovee et al., 2004) [13]
	*Arxula adeninivorans*	Quantification (2 ng/L)	Amperometry or biochemistry (phyK)	(Pham et al., 2013) [14]
Androgenic and Anti-androgenic compounds	*Saccharomyces cerevisiae*	Quantification (15 nM for Testosterone)	Two-hybrids System, (LacZ).	(Lee et al, 2003) [15]
	*Saccharomyces cerevisiae*	Quantification (5 nM)	Two-hybrids System, (GFP).	(Ogawa et al., 2010) [16]
Glucocorticoids (cortisol, corticosterone)	*Arxula adeninivorans*	Quantification (0.3 μM)	Amperometry or biochemistry (phyK)	(Pham et al., 2016) [17]
**Pharmaceuticals** (omeprazole, lansoprazole)	*Arxula adeninivorans*	Quantification (95 μg/L)	Amperometry or biochemistry (phyK)	(Pham et al., 2015) [18]
**Mycotoxins:**				
T-2 toxin and other trichothecenes such as verrucarin A	*Kluyveromyces fragilis*	Yes/No	Growth inhibition (disk halo)	(Schappert and Khachatourians, 1984) [19]
Trichothecene mycotoxins	*Kluyveromyces marxianus*	Quantification (1 pg/L)	Colorimetry (LacZ)	(Engler et al., 1999) [20]
Mycotoxin Zearalenone, and other compounds with estrogenic activity	*Saccharomyces cerevisiae*	Quantification (1 μg/L)	Metabolic construct	(Mitterbauer et al., 2003) [21]
**Heavy metals:**				
	*Saccharomyces cerevisiae*	Quantification (0.5 mM Cu^2+^)	Amperometry (LacZ).	(Lehmann et al., 2000) [22]
	*Saccharomyces cerevisiae*	Quantification (5 × 10^-7^M Cu^2+^)	Fluorescence (GFP)	(Shetty et al., 2004) [23]
	*Saccharomyces cerevisiae*	Quantification (5 × 10^-7^M Cu^2+^)	Luminescnce (Luc)	(Roda et al., 2011) [24]
	*Saccharomyces cerevisiae*	Quantification (1 μM Cu^2+^)	Colorimetry (ADE2)	(Vopálenská et al., 2015) [25]
Cadnium, Arsenic.	*Hansenula polymorpha*	Quantification (1 mM Cd)	Fluorescence (GFP)	(Park et al., 2007) [26]
**Marine toxins:**				
Okadaic acid, pectenotoxin-11, portimine	*Saccharomyces cerevisiae*	Quantification (19 nM OA)	Colorimetry (LacZ)	(Richter and Fidler, 2015) [27]
Ciguatoxins	*Saccharomyces cerevisiae*	Quantification (0.1 ng/L PCTX3C)	Colorimetry or fluorescence (LacZ)	(Martin-Yken et al., 2018) [28]
**Biological Organics (BOD measure):**	*Trichosporon cutaneum*	Quantification (3 mg/L)	Amperometry	(Hikuma et al., 1979) [29]
	*Arxula adeninivorans* and other yeast species	Quantification (2.4 mg/L)	Cellular growth	(Yudina et al., 2015) [30]
	*Saccharomyces cerevisiae*	Quantification (2 mg/L)	Spectrophotometry	(Nakamura, 2007) [31]
**Medical Domain:**				
**Pathogens** (of any type)	*Saccharomyces cerevisiae*	Quantification (nM range)	SPR (antigen cell surface display)	(Venkatesh et al., 2015) [32]
**Fungal pathogens**	*Saccharomyces cerevisiae*	Yes/No (nM range)	Colorimetry, Engineered GPCR	(Ostrov et al., 2017) [33]
**Carcinogens, genotoxics:**	*Saccharomyces cerevisiae*	Quantification (mg/mL range)	Reversion frequency (DEL assay)	(Brennan and Schiestl, 1998, 2004) [34,35]
	*Saccharomyces cerevisiae*	Quantification (variable)	Fluorescence (GFP)	(Benton et al., 2007) [36]
Pro-carcinogens	*Saccharomyces cerevisiae*	Quantification (μg/mL range)	CPR-CYP and RAD54-GFP expression	(Bui et al., 2016) [37]
PI3K inhibitors (oncogenesis related screen)	*Saccharomyces cerevisiae*	Quantification (μM range)	Reconstituted PI3K pathway	(Fernández-Acero et al., 2012) [38]
**Screens:**				
for Matrix Metalloproteinases (MMPs) inhibitors (anticancer)	*Pichia pastoris*	Quantification (nM range)	Cell surface expression	(Diehl et al., 2011) [39]
for Anti-Malarial Compounds with artemisinin-like activities	*Saccharomyces cerevisiae*	Yes/No (μM range)	Growth inhibition	(Mohamad et al., 2012) [40]
for Inhibitors of Human Cytomegalovirus Protease	*Saccharomyces cerevisiae*	Quantification (μM range)	Target-specific HTS system	(Cottier et al., 2006) [41]

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
