# Peer review of "Yeast-Based Biosensors: Current Applications and New Developments"

_biosensors, 2020, doi:10.3390/bios10050051_

Round 1

Reviewer 1 Report

Peer review report for Yeast Based Biosensors: current application and new developments

The author provided an overview review of current applications for yeast-based biosensors and their future prospects. I believe that this manuscript looks well-written with enough number of references. So, I recommend an editor publish this manuscript in MDPI biosensors after several minor revisions.

I listed the questions and comments below.

1. I believe that Table 1 can be improved by adding another information. For example, types of detection (yes/no or quantification), limits of quantification and etc.

2. If possible, I would like to read about the immobilization of yeast to transducers written by the author. I have been studying about enzyme-based biosensors. In enzymatic biosensors, the immobilization method affects sensor characteristics such as sensitivity, response time, sensors life-time, selectivity. and etc. Is it the same in yeast-based biosensors?

3. There are review papers about yeast-based biosensors that published in the last five years (A. Adeniran et al., Yeast-based biosensors: design and applications, FEMS Yeast Research, 15, 1-15, 2015, S. Jarque et al., Yeast Biosensors for Detection of Environmental Pollutants: Current State and Limitations, Trends in Biochemistry, 34, 408-419, 2016). Please point out the difference between your review and the above papers. 

Author Response

Peer review report 1 for “Yeast Based Biosensors: current application and new developments”

The author provided an overview review of current applications for yeast-based biosensors and their future prospects. I believe that this manuscript looks well-written with enough number of references. So, I recommend an editor publish this manuscript in MDPI biosensors after several minor revisions.

I listed the questions and comments below.

  1. I believe that Table 1 can be improved by adding another information. For example, types of detection (yes/no or quantification), limits of quantification and etc.

To address this concern, I have added two columns in Table 1. One of them describes shortly the sensing and detection system, while the other indicates the type of detection (« Yes/No » or « Quantification ») as well as the limit of detection (LoD) or EC50 when these informations were available. However, regarding these quantitative values, they have to be taken with caution since a given biosensor can have a certain level of sensitivity (Limit of Quantification or Limit of Detection) for one compound in given conditions and a different one for other compounds, or in different conditions. This is indeed generally the case, as most of these biosensors are designed to detect multiple molecules. Hence I have chosen the values for Table 1 by considering the compound of refrence when there was one in the section (exemple : Copper for the heavy metals, cortisol for the endocrine disruptors). But the readers actually interested in these precise numeric values should still refer to the original publications that are cited in the table and in the references, where they will find accurate and precise sensitivity informations for the actual molecule they are interested in.

  1. If possible, I would like to read about the immobilization of yeast to transducers written by the author. I have been studying about enzyme-based biosensors. In enzymatic biosensors, the immobilization method affects sensor characteristics such as sensitivity, response time, sensor’s life-time, selectivity. and etc. Is it the same in yeast-based biosensors?

Indeed, the immobilization strategies used by scientists to developp yeast biosensors have a notable influence on the biosensor characteristics, including of course their storage time and their sensitivity. For example, yeast cells can be lyophilized, integrated in paper or freeze-dried and stay viable at -80°C for several months. They can be entrapped in biocompatible matrices based on various polymers, alginate, silicca gels and there again kept for long periods at low temperature. Such immobilized yeast cells are obviously easier to handle than yeast cells in solution ; however the use of polymers as support is not without consequences since they limit the diffusion of the molecules of interest and hence can lower significantly the sensitivity of immobilized cells. Finally, the rise of nanotechnologies and particularly microfluidics is changing the game : it is now possible to isolate yeast cells in nano-cages and expose them individually to the samples to analyse. To conclude, the question of the cells immobilization method in yeast based biosensors could be the subject of a review in itself, and is in addition a rapidly evolving domain.  

  1. There are review papers about yeast-based biosensors that published in the last five years (A. Adeniran et al., Yeast-based biosensors: design and applications, FEMS Yeast Research, 15, 1-15, 2015, S. Jarque et al., Yeast Biosensors for Detection of Environmental Pollutants: Current State and Limitations, Trends in Biochemistry, 34, 408-419, 2016). Please point out the difference between your review and the above papers.

              The review by Adeniran and colleagues published in 2015 in FEMS Yeast Research is an excellent and very detailled study on yeasts sensing pathways engineered by scientists to detect different molecules. It really focuses in depth on the molecular and cellular biology mechnisms of S. cerevisiae and is thus in my opinion interesting mostly for yeast researchers. I believe the reading of their work could be relatively difficult for scientists non familiar with S. cerevsiae molecular biology. On the other hand, I have tried to make the submitted manuscript as clear as possible and understandable by anyone interested in biosensors even without a microbiology background, including hopefully non – biologists scientists. Regarding the remarkable review by Jarque and colleagues published in Trends in Biochemistry, it deals with environmental pollutants only, while I have tried here to cover all domains where yeast biosensors can be applicable. In addition, their work  was also clearly written with the optics of underlining the limitations of the use of yeast based biosensors circa 2015-2016, while the submitted manuscript attempts on the opposite to describe the future trends and the potential developments of these biosensors in the years to come. To conclude, as impressive as these two other reviews are, I think that our three works are complementary and not redondant.

Reviewer 2 Report

I thoroughly looked into the manuscript. This paper reviewed yeast-based biosensors in the areas of environmental and medical/health domains. It is well-written and narrows the scope of the work with current applications and new developments. I added a couple of comments to improve the quality of the manuscript. I would recommend to accept this paper in the journal of Biosensor after revisions.

Line 80: What are the acronyms of PDR and ABC?

Line 87: I recommend the author to add/summarize "yeasted based sensing technology" section before “Current applications” as sensing techniques are important in biosensors. This section will provide a smooth transition “Introduction” to “Application”.

Table 1: Also, here I would suggest the author to add detection system (method) in another column. The author already provides some of the references (luminescence, colorimetry… and so on). It will be clear for readers.

Table 1: The references are mostly old. I would suggest the author to add up-to-data references, 2016 and 2019 years.

Line 96: Please, add space “Table 1”.

Line 340: This section covers current advances and future development. So the title would be better to use “Current and Future Developments”.

It would be appreciated the readers if the author adds a couple of sentences in 3.3 (or 3) about medical domain technological developments since it is not mentioned in the section. This will strengthen the manuscript.

Author Response

Peer review report 2

I thoroughly looked into the manuscript. This paper reviewed yeast-based biosensors in the areas of environmental and medical/health domains. It is well-written and narrows the scope of the work with current applications and new developments. I added a couple of comments to improve the quality of the manuscript. I would recommend to accept this paper in the journal of Biosensor after revisions.

Line 80: What are the acronyms of PDR and ABC?

« PDR » stands for Pleiotropic Drug Resistance and « ABC » for ATP-Binding Cassette. In S. cerevisiae, the PDR genes encode a family of multidrug resistance ATP binding cassette (ABC) proteins. I appologize for using these acronyms without giving the full names in the first manuscript version, and I have corrected this in the revised version, Lines 81-82.

Line 87: I recommend the author to add/summarize "yeasted based sensing technology" section before “Current applications” as sensing techniques are important in biosensors. This section will provide a smooth transition “Introduction” to “Application”.

I have added a section as suggested.

Lines 88-102 : « Yeast based sensing technology has thus evolved from using the natural potential of yeast cells such as their sensitivity to toxic molecules or their ability to metabolize organic coumponds and simply following their growth, towards the design of more and more complex genetically modified strains. Notably, many biosensors have been constructed by integrating in yeast cells heterologous genes conferring them new recognition capabilities. These exogenous sensors proteins can be coupled directly or indirectly to transcription factors which in turn activate a reporter gene, either metabolic or driving a signal which can be easily followed by colorimetry, fluorescence, luminescence, amperometry, etc. Such approaches have been used by yeast scientists worldwide to design biosensors for a very wide range of applications (see below, part 2.). But several other smart sensing mechnisms have also been developed for specific purposes, like for example using the yeast genetic recombination frequency to assess the presence of genotoxic compounds. In addition, yeast based sensing technology is a field of constant evolution. Increasingly sophisticated mechanisms are currently being designed. Moreover, the rise of synthetic biology combined to computer assisted structural biology are opening exciting future prospects (see below, part 3.). »

Table 1: Also, here I would suggest the author to add detection system (method) in another column. The author already provides some of the references (luminescence, colorimetry… and so on). It will be clear for readers.

As suggested by reviewer 2, I have added a column in Table 1 to describes shortly the sensing and detection system.

Table 1: The references are mostly old. I would suggest the author to add up-to-data references, 2016 and 2019 years.

I have changed some of the oldest references of Table 1 for updatded ones. However, I would also like to keep some others of these refrerences, since I consider important to acknowledge the initial developpers of certain original yeast based biosensors described in this table.
I have also added three recent references in the MS :

Aronoff-Spencer et al.,  Biosensors and Bioelectronics 2016

Hernández-Neuta et al., J. Intern. Med. 2019

Castano-Cerezo et al., Frontiers in Bioengineering and Biotechnology. 2020.

Line 96: Please, add space “Table 1”.

I have added the space and completed the legend of the table to take into account the new data presented. Lines 108-117.

Line 340: This section covers current advances and future development. So the title would be better to use “Current and Future Developments”.

I have made the corresponding change, Line 362.

It would be appreciated the readers if the author adds a couple of sentences in 3.3 (or 3) about medical domain technological developments since it is not mentioned in the section. This will strengthen the manuscript.

I added two sentences and two new recent relevant references : Aronoff-Spencer et al.,  Biosensors and Bioelectronics 2016 and Hernández-Neuta et al., J. Intern. Med. 2019.

« These technogical developments in the detection and and results analysis are regarded by the health care providers as promising perspectives towards simplifying, standardizing and automating biosensors-based diagnostic techniques, as they allow combining high precision and sensitivity with the connectivity and computational power of smartphones. Hence, clinical smartphone diagnostics methods are a rapidly emerging field whose area of applications ranges from hematology to rapid infectious disease diagnostics and digital pathology [88]. » Line 445-451.

« The same versatile bio-brick approach has also been developed into an integrated diagnostic platform that combines sensing of Hepatitis C virus core antigen with a connected signal acquisition/processing through a smartphone-based potentiostat [90]. » line 462-464.

Reviewer 3 Report

This review surveys the developments of  the yeast based biosensors for the applications in environmental and medical domains. It presents some of the technological developments currently undertaken by academic and corporate scientists to further drive yeasts biosensors. This is an interesting work but with some flaws. A minor revision is needed before consider to publish. My comments are listed below.

  1. As a review work in biosensor, a good schematic illustration is better than hundreds of words for the author to understand the work principle of a biosensor.
  2. The sensitivity is a key parameter of a biosensor, but such information is ignored.

Author Response

Peer review report 3

This review surveys the developments of  the yeast based biosensors for the applications in environmental and medical domains. It presents some of the technological developments currently undertaken by academic and corporate scientists to further drive yeasts biosensors. This is an interesting work but with some flaws. A minor revision is needed before consider to publish. My comments are listed below.

 1)   As a review work in biosensor, a good schematic illustration is better than hundreds of words for the author to understand the work principle of a biosensor.

Absolutely. Figure 1 of the submitted manuscript,  called « General scheme of a yeast biosensor purpose and functioning » is actually a schematic illustration of a biosensor’work principle.

2)    The sensitivity is a key parameter of a biosensor, but such information is ignored.

Indeed, I agree with reviewer 3 and I have been providing this information for some of the biosensors mentionned in the manuscript. However, it is impossible to give these numbers for all examples described in this revue for two main reasons. First, biosensors generally can have a certain level of sensitivity (Limit of Quantification or Limit of Detection) for one compound in given conditions and a different one for another (or other) compound(s) in other conditions. It would be extremely heavy to give all these numeric data, and this is why I did not put this information everywhere. I do believe that the readers interested in having these precise sensitivity informations will refer to the original publications that are cited in the references. Second, I have tried my best to keep this revue an easy and - if possible - nice read, and too many quantitative concentrations informations would certainly not allow to keep it that way.   So, to address this concern without harming the manuscript reading, I have added in Table 1 a column which indicates the limit of detection (LoD) or EC50 when these informations were available. However, these quantitative values have to be taken with caution for the reasons that I have explained above (different sensitivity for different compounds or conditions). I have chosen the values for Table 1 by considering the compound of reference when there was one in the section (exemple : Copper for the heavy metals, cortisol for the endocrine disruptors).
